# Ultrahigh transverse thermoelectric power factor in flexible Weyl semimetal WTe$_2$

Yu Pan [1✉], Bin He[1], Toni Helm [2], Dong Chen [1], Walter Schnelle[1] & Claudia Felser [1✉]

Topological semimetals are well known for their interesting physical properties, while their mechanical properties have rarely received attention. With the increasing demand for flexible electronics, we explore the great potential of the van der Waals bonded Weyl semimetal WTe$_2$ for flexible thermoelectric applications. We find that WTe$_2$ single crystals have an ultrahigh Nernst power factor of ~3 Wm$^{-1}$K$^{-2}$, which outperforms the conventional Seebeck power factors of the state-of-the-art thermoelectric semiconductors by 2–3 orders of magnitude. A unique band structure that hosts compensated electrons and holes with extremely high mobilities is the primary mechanism for this huge Nernst power factor. Moreover, a large Ettingshausen signal of ~5 × 10$^{-5}$ KA$^{-1}$m is observed at 23.1 K and 9 T. In this work, the combination of the exceptional Nernst–Ettingshausen performance and excellent mechanical transformative ability of WTe$_2$ would be instructive for flexible micro-/nano-thermoelectric devices.

[1] Max Planck Institute for Chemical Physics of Solids, Nöthnitzer Str. 40, Dresden 01187, Germany. [2] Dresden High Magnetic Field Laboratory (HLD-EMFL), Helmholtz-Zentrum Dresden-Rossendorf, Dresden 01328, Germany. ✉email: Yu.Pan@cpfs.mpg.de; Claudia.Felser@cpfs.mpg.de

Topological semimetals are gaining increasing attention for various frontier applications including quantum computing, spintronics, and thermoelectrics, owing to their unique physical transport properties and topological states. By converting heat into electricity and vice versa in a solid state, thermoelectric technology holds great potential for power generation and solid-state cooling[1,2]. With no moving parts, thermoelectric devices avoid vibration, noise, and fatigue problems, making them highly suitable for cooling and precise temperature control of laser diodes, medical storage systems, and infrared detectors[3–5]. Recently, topological semimetals have shown large anomalous Nernst signals[6,7]. Compared with the longitudinal Seebeck–Peltier effects, transverse thermoelectric applications based on Nernst–Ettingshausen effects require only one material instead of coupled p- and n-type legs, making device construction considerably simpler[8,9]. Moreover, the output of a Nernst–Ettingshausen cooler depends on its geometry, making them suitable for use in micro- or nanodevices[8,9].

In addition to miniaturization and high thermoelectric performance, the mechanical flexibility of thermoelectric devices has become increasingly important in recent years and has driven the development of novel materials and composites[10–12]. To date, the studies of flexible thermoelectrics have mainly focused on the longitudinal Seebeck effect. In contrast, flexible micro- or nanoscale Nernst–Ettingshausen coolers have rarely been investigated. One of the main challenges for flexible longitudinal and transverse thermoelectrics, is that most inorganic thermoelectric materials are inherently rigid. Present flexible Seebeck effects rely on conductive polymers such as poly(3,4-ethylenedioxythiophene): poly(styrenesulfonate) (PEDOT:PSS)[13], embed conducting polymers to form organic/inorganic hybrids[14], and inorganic films (e.g., classic thermoelectric compounds such as $Bi_2Te_3$ and $Sb_2Te_3$) grown on flexible substrates[15]. However, these strategies have certain disadvantages, including the low electrical conductivity of the polymer and the cumbersome process for producing thin films and hybrid materials, which induce critical device-level limitations for practical applications. $Ag_2X$ (X = Te, Se, S) semiconductors are rare examples of material that have the high-longitudinal thermoelectric performance as an inorganic material and the mechanical flexibility like organic materials[16]. Meanwhile, the development of flexible transverse thermoelectrics is still lacking, particularly based on inorganic materials. Thus, exploring materials with combined high transverse thermoelectric performance and mechanical flexibility is highly desired. Van der Waals bonded topological semimetals are considered potential candidates for achieving flexible nano-/ microscale Nernst–Ettingshausen coolers.

In this work, layered $WTe_2$ single crystals are adopted as an example to demonstrate topological semimetals as ideal candidates for flexible transverse thermoelectric applications. Owing to the van der Waals bonding between the Te layers, $WTe_2$ has excellent flexibility and can be twisted into different shapes. Furthermore, as a consequence of the extremely high charge-carrier mobilities and semimetallic transport behavior, large unsaturated Nernst and Ettingshausen signals with maximum values of ~7000 $\mu VK^{-1}$ (at 11.3 K and 9 T) and ~5 × $10^{-5}$ $KA^{-1}m$ (at 23.1 K and 9 T), respectively, are observed. Most importantly, an ultrahigh Nernst power factor of ~3 $Wm^{-1}K^{-2}$ is achieved, which is 2–3 orders higher than the Seebeck power factors typically reported. The present work highlights the role of flexible topological semimetals for thermoelectric cooling, particularly as flexible devices, in niche applications.

## Results

**Suitability of $WTe_2$ for flexible transverse thermoelectrics.** The semimetal $WTe_2$ crystallizes in an orthorhombic structure with a space group of $Pmn2_1$[17]. The $a$-axis has the smallest lattice constant, and the Te–Te layers in the $ab$-plane are bonded along the $c$-axis by van der Waals forces (Fig. 1a). Therefore, $WTe_2$ forms ribbon-shaped single crystals mainly growing along the $a$-axis, with the $c$-axis perpendicular to the layers (Fig. 1b and Supplementary Fig. 1). Owing to the weak van der Waals bonds between layers, the crystals can be easily exfoliated into thin sheets. Moreover, $WTe_2$ single crystals have excellent flexibility that enables reversible strong deformations. As an example, Fig. 1b shows single crystals that are artificially deformed/buckled into letters to spell out "$WTe_2$," and into a Möbius band, as shown in the inset. The high flexibility and two-dimensional van der Waals bonding characteristics make $WTe_2$ single crystals ideal for flexible device applications, particularly at low dimensions.

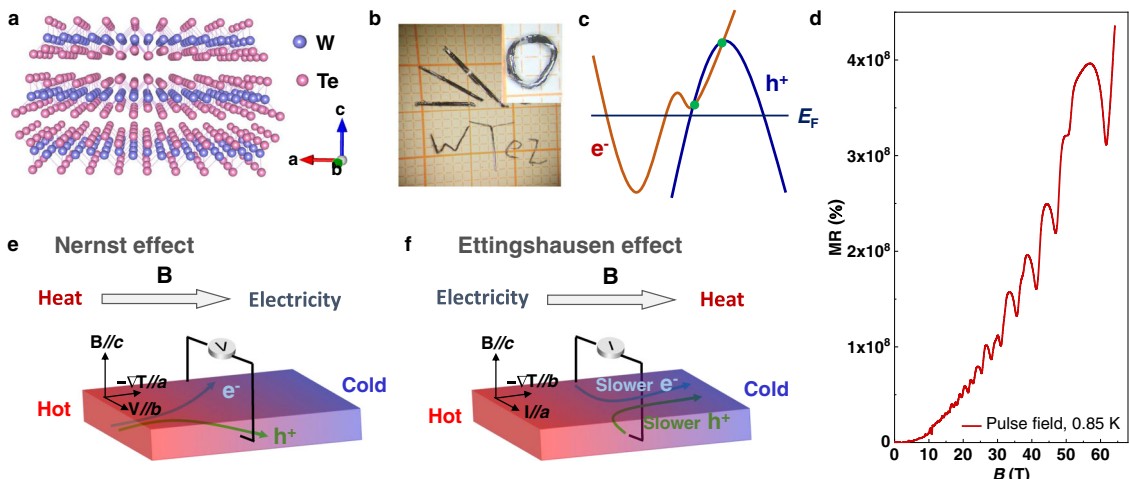

**Fig. 1 $WTe_2$ basic properties and the Nernst–Ettingshausen effect. a** Crystal structure of $WTe_2$. Te–Te layers in the $ab$-plane are bonded by van der Waals forces and the crystals can be cleaved easily. **b** Photograph of $WTe_2$ single crystals, and the spelling of "$WTe_2$" using artificially deformed crystals. Inset shows a Möbius strip formed from a $WTe_2$ single crystal. **c** Schematic of the band structure of $WTe_2$ Weyl semimetal. Electrons and holes show nearly perfect compensation at the Fermi energy. The green dots show a pair of Weyl points, which are above the Fermi energy. **d** Magnetoresistance of the studied $WTe_2$ single crystal at 0.85 K recorded using a magnetic-field pulse up to 65 T. **e** Schematic illustrations of Nernst effect. **f** Ettingshausen effect. The orientation of the experimental setup corresponding to the crystal axes is shown. Both the Nernst and Ettingshausen devices need only one material, and therefore have lower complexity than Seebeck devices.

WTe$_2$ single crystals have been extensively investigated in condensed matter physics and materials science fields[17–20]. To date, WTe$_2$ is recognized as a type-II Weyl semimetal with an almost equal number of electrons and holes. Fig. 1c shows the compensation behavior of electrons and holes at the Fermi energy, which leads to extremely large magnetoresistance; and the Weyl points are ~50 meV above the Fermi energy[21,22]. Fig. 1d and Supplementary Fig. 2 show the giant magnetoresistance and clear magnetic quantum oscillations for the studied WTe$_2$ single crystal. The strong oscillation amplitude and large residual resistance ratio (RRR) of 1572 (Supplementary Fig. 3) demonstrate the high crystal quality. Considering the intrinsic semimetallic character and high mobility of both electrons and holes, WTe$_2$ has ideal preconditions for achieving a strong transverse thermoelectric effect. However, studies on the transverse thermoelectric performance of WTe$_2$ single crystals from the point-of-view of thermoelectric applications are rare[23,24]. To our best knowledge, no prior study has discussed the Ettingshausen effect for this compound.

Compared with the more common longitudinal thermoelectric devices based on the Seebeck effect, a transverse thermoelectric device using Nernst and Ettingshausen effects can simplify the module fabrication considerably by avoiding the need for assembled pairs[9,25]. Ettingshausen coolers have been investigated actively in the 1960s[26,27], but the requirement of a large external magnetic field limits their practical application. Therefore, a great breakthrough of the Nernst–Ettingshausen performance particularly under a relatively small magnetic field, is critical for future niche applications, e.g., where vibration-free operation is more important than the cost of providing a strong magnetic field. The advantages of permanent magnets should also be considered. As shown in Fig. 1e, f, the Nernst effect can convert heat into electricity, and the Ettingshausen effect converts the electricity to heat. In the presence of a magnetic field, electrons and holes synergistically contribute to the Nernst and Ettingshausen effects, for which semimetals with two types of charge carriers are highly advantageous. Considering the benefits of device designs based on a single material with high-mobility charge carriers, compensated electrons and holes, and the high flexibility of WTe$_2$, we foresee an inspiring potential for WTe$_2$ in transverse thermoelectric applications at the micro- and nanoscales, particularly for applications where mechanical flexibility is required.

**Large Nernst and Ettingshausen signals in WTe$_2$ single crystals.** The studied WTe$_2$ single crystals yield large Nernst and Ettingshausen signals at low temperatures with no saturation up to a magnetic field $B = 9$ T. As shown in Fig. 2a, the Nernst signal $S_{yx}$ increases as a function of the magnetic field, and the maximum value reaches 7000 μVK$^{-1}$ at 11.3 K and 9 T. Similarly, as shown in Fig. 2b, the Ettingshausen signal also increases as a function of the magnetic field, with a maximum of $5 \times 10^{-5}$ KA$^{-1}$m at 23.1 K and 9 T. For a semimetal with two types of charge carriers, the Nernst coefficient $N$ ($N = S_{yx}/B$) is determined by:

$$N = \frac{(N_e\sigma_e + N_h\sigma_h)(\sigma_e + \sigma_h) + (N_e\sigma_e\mu_h - N_h\sigma_h\mu_e)(\sigma_e\mu_h - \sigma_h\mu_e)B^2 + \sigma_e\sigma_h(\mu_h + \mu_e)(\alpha_h - \alpha_e)}{(\sigma_e + \sigma_h)^2 + (\sigma_e\mu_h - \sigma_h\mu_e)^2 B^2}$$

(1)

where $N_e$ and $N_h$, $\sigma_e$ and $\sigma_h$, $\mu_e$ and $\mu_h$, $\alpha_e$ and $\alpha_h$ denote the Nernst coefficient, conductivity, mobility, and Seebeck coefficient of electrons and holes, respectively, $e$ is the elementary charge, and $B$ is the magnetic field[28]. Accordingly, the high mobilities of both electrons and holes can dramatically increase $N$. Moreover, with comparable mobility and concentration of the two charge carrier types, as well as a magnetic-field-independent $N$ (i.e. linear Nernst thermopower as a function of $B$), the terms with $B^2$ in Eq. (1) should have a minor influence on the total $N$. Therefore, in the

case of WTe$_2$ we approximated $N$ as:

$$N \approx \frac{(N_e\sigma_e + N_h\sigma_h)(\sigma_e + \sigma_h) + \sigma_e\sigma_h(\mu_h + \mu_e)(\alpha_h - \alpha_e)}{(\sigma_e + \sigma_h)^2}$$

(2)

Consequently, the high charge-carrier mobilities are the main source of large Nernst thermopower of WTe$_2$. Most importantly, the simultaneous observation of large Nernst and Ettingshausen signals and the mechanical flexibility of WTe$_2$ makes it a unique candidate for flexible transverse thermoelectric generators, which have never been achieved using other high-Nernst-performance topological semimetals, such as the state-of-the-art Bi[29,30], and the Weyl semimetal NbP[8].

The magnetic field responses of both Nernst and Ettingshausen signals decrease at elevated temperatures, which is also the case for the Seebeck coefficient (Fig. 2c). In general, the magnetic-field dependence of the Seebeck coefficient is stronger at lower temperatures (i.e., below 10 K) and becomes much weaker at higher temperatures. The temperature dependence is further shown in Fig. 2d–f. While the Nernst signal peaks between 10 K and 15 K, the Ettingshausen signal monotonically decreases from 20 K to 100 K. The maximum Nernst signal indicates that the Fermi energy is located near the electron-hole compensation point at this temperature range. It is challenging to observe the Ettingshausen signal below 20 K, since the sample geometry and high thermal conductivity of WTe$_2$ require a large electrical current to create an effective transverse temperature gradient. The relatively large current-induced Joule heating results in a large temperature drift that prevented us from reaching below 20 K. Nevertheless, we expect that the Ettingshausen signal further increases at 10–15 K, at which the maximum Nernst signal is generated. Most likely, this is the temperature range where the electrons and holes compensate each other most efficiently. In addition, a rough estimation of the Ettingshausen coefficient can be obtained from the Nernst coefficient based on the Onsager relation, which can be found in the Supplementary Information.

It is noteworthy that, at 0 T, the Seebeck coefficient (Fig. 2f) changes from positive to negative with increasing temperature with a maximum value at 10 K. The turning point can be understood considering two aspects. First, phonon scattering sharply increases above 10 K, leading to the phonon drag effect on the turning point of the Seebeck coefficient. Second, it may indicate that the electrons start to play a significant role in determining the Seebeck coefficient. Accordingly, the changeover temperature for the significant competition between electrons and holes is proposed to be near 10 K. As a semimetal with two types of charge carriers, the total Seebeck coefficient $\alpha$ of WTe$_2$ is described as follows[31].

$$\alpha = (\alpha_e\sigma_e + \alpha_h\sigma_h)/(\sigma_e + \sigma_h)$$

(3)

Below 50 K, holes contribute more to $\alpha$, i.e., $|\alpha_h\sigma_h| > |\alpha_e\sigma_e|$, thus resulting in a positive $\alpha$ value. Nevertheless, the contribution from electrons starts to play a significant role above 10 K, leading to the negative slope in $\alpha$ at higher temperatures.

**Band features and charge carrier concentration in WTe$_2$.** The exact concentration and mobility of electrons and holes, as well as the effective masses are adopted to further reveal the thermoelectric responses. In a system with two types of charge carrier, the Hall charge-carrier concentration and mobility can be resolved by fitting the Hall conductivity $\sigma_{xy}$[32].

$$\sigma_{xy} = \frac{\rho_{yx}}{\rho_{xx}^2 + \rho_{yx}^2} = \left[ \frac{-n_e\mu_e^2}{1 + (\mu_e B)^2} + \frac{n_h\mu_h^2}{1 + (\mu_h B)^2} \right] eB$$

(4)

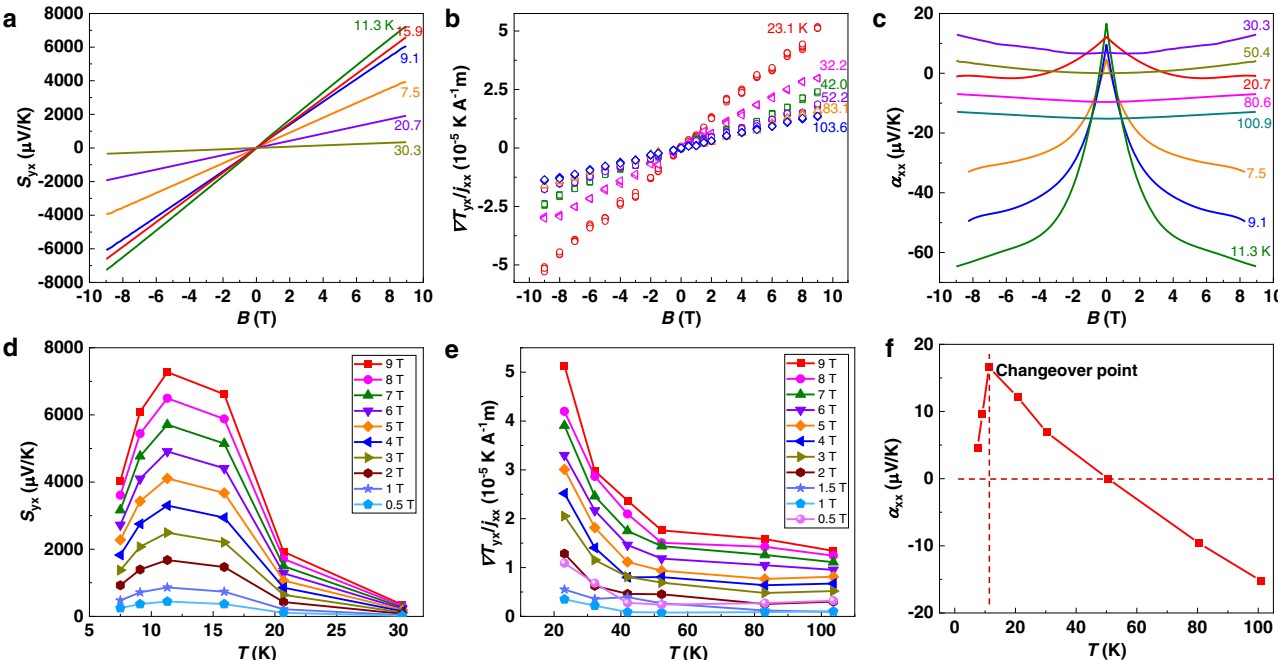

**Fig. 2 Nernst, Ettingshausen, and Seebeck effect in WTe₂. a** Nernst signal from 7.5 K to 30.3 K. **b** Ettingshausen signal from 23.1 K to 103.6 K. **c** Seebeck coefficient from 7.5 K to 100.9 K. **d** Nernst signal as a function of temperature. **e** Ettingshausen signal as a function of temperature. **f** Temperature-dependent Seebeck coefficient at 0 T of a WTe₂ single crystal. Both the Nernst and Ettingshausen effects show large, non-saturating signals up to 9 T. The maximum Nernst effect is observed in the range of 10–15 K, and the maximum Ettingshausen effect observed in the present work was at 23.1 K, which is the lower temperature limit of our measurements.

where $\rho_{yx}$ and $\rho_{xx}$ are the Hall and longitudinal resistivity ($\rho_{xx}$ and $\rho_{yx}$ are shown in Fig. 3a, b), and $n_e$ and $n_h$ are the charge carrier concentration of electrons and holes, respectively. Detailed fitting results can be found is Supplementary Fig. 4, and the obtained parameters are shown in Fig. 3c, d. Both the electron and hole concentrations increase with temperature from 2 to 50 K, yielding similar trends. The electron and hole concentrations are very similar, with an order of $10^{19}$ cm$^{-3}$; this was also confirmed by the estimated values from the oscillation (Supplemental Table 1). The ratio of the electron and hole concentrations ($n_e/n_h$) is shown in the inset of Fig. 3c. Its value is very close to unity in the entire temperature range from 2 to 50 K, thus demonstrating the perfect compensation of the electrons and holes over a wide temperature range. In contrast, the mobility of the electrons is slightly higher than that of the holes, particularly at low temperatures below 10 K (Fig. 3d). The ratio of the electron and hole mobility ($\mu_e/\mu_h$) is shown in the inset of Fig. 3d. This ratio decreases rapidly from 2 to 10 K, and then saturates to a constant $\mu_e/\mu_h \approx 3$ at approximately 30 K. The saturated decreasing trend of $\mu_e/\mu_h$ at 10 K indicates again that the changeover temperature to achieve the most effective competition between electron and hole conduction is at 10 K, which is consistent with the results of the Seebeck analysis. This may explain why the maximum $S_{yx}$ is achieved at 10 K.

Furthermore, we propose that the holes have a higher Seebeck coefficient $\alpha_h$ than the electrons $|\alpha_e|$, which can be explained based on the effective masses of the charge carriers. Experimentally, the effective masses of the electron and hole pockets are revealed in the analysis of Shubnikov–de Haas (SdH) oscillations of the longitudinal resistivity. Both $\rho_{xx}$ and $\rho_{yx}$ show clear quantum oscillations below 10 K at higher magnetic fields above 5 T. The fast Fourier transformation (FFT) analysis of the oscillatory part reveals four main frequencies, associated with four Fermi pockets including two electron pockets and two-hole pockets, labeled e1 and e2, and h1 and h2, respectively in Fig. 3e.

The four frequencies $F = 94$ T (h1), 130 T (e1), 144 T (e2), and 159 T (h2), are very close to the previously reported values[33,34]. We extracted the effective cyclotron mass $m^*$ values from fitting the temperature dependence of the FFT amplitudes $A_{FFT}$, according to the Lifshitz–Kosevich formula below[35]

$$A_{FFT} = A_0 \frac{2\pi^2 k_B m^* T/e\hbar\bar{B}}{\sinh\left(2\pi^2 k_B m^* T/e\hbar\bar{B}\right)}, \quad \frac{1}{\bar{B}} = \frac{1}{B_1} + \frac{1}{B_2} \quad (5)$$

Here, $A_0$ is a constant, $k_B$ is Boltzmann's constant, $T$ is the temperature, $\hbar$ is the reduced Planck constant, and $B_1$ and $B_2$ are the starting and ending magnetic fields of the FFT field window, respectively. As shown in Fig. 3f, the effective masses of h1, e1, e2, and h2 are resolved as $0.33m_0$, $0.26m_0$, $0.29m_0$, and $0.31m_0$, respectively, in units of the free electron mass $m_0$. Generally, the effective masses of the hole pockets are slightly higher than those of the electron pockets, leading to a larger $\alpha_h$ than $\alpha_e$. This may be one of the reasons for the positive $\alpha$ at temperatures below 10 K and the higher $\mu_e/\mu_h$ ratio.

**Thermoelectric performance and potential application of WTe₂.** Owing to the sharp linear-band dispersions, topological semimetals such as WTe₂ exhibit high mobility and even compensated electron–hole conduction, making them excellent candidates for transverse thermoelectric applications. Fig. 4a compares the mobility and Nernst thermopower of a few topological materials, including WTe₂, ZrTe₅[36], NbP[8], and Cd₃As₂[37], as well as Bi single crystals[29,30]. The Nernst signal strength has a positive correlation with the charge-carrier mobility. This is the reason why topological semimetals, which usually have low effective mass and even massless Fermions, are highly promising for achieving high Nernst signals. Furthermore, the compensated conduction of the electrons and holes is essential for achieving large Nernst thermopowers. Therefore, the critical Fermi energy at which the electrons and holes compensate most effectively also strongly affects the Nernst signal. For example, WTe₂ and NbP[8]

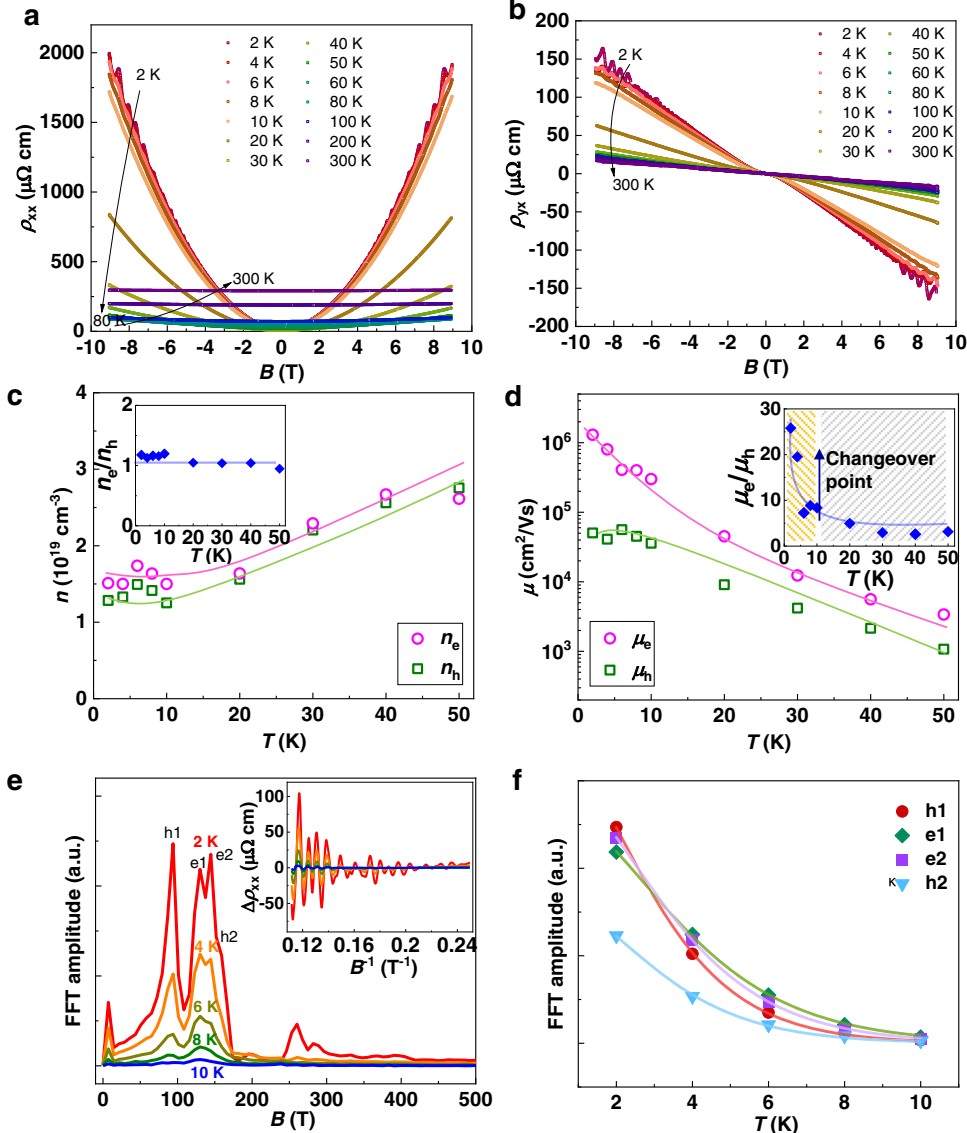

**Fig. 3 Charge-carrier mobilities and effective cyclotron masses of WTe$_2$. a** Longitudinal resistivity as a function of magnetic field. **b** Hall resistivity as a function of magnetic field. **c** Carrier concentration. **d** Mobility of both electrons and holes. The insets in (**c**) and (**d**) present the ratios of the concentration and mobility of electrons to holes, respectively. **e** Fast Fourier transform (FFT) spectra of the oscillatory part of $\rho_{xx}$ at different temperatures from 2 K to 10 K. The inset shows the background-subtracted oscillations as a function of $1/B$. **f** Temperature dependence of the FFT amplitudes for the h1, e1, e2, and h2 Fermi pockets. The points are experimental values and the lines are fits according to the Lifshitz–Kosevich formula[35].

have their maximum Nernst thermopowers at 10 K and 109 K, respectively, corresponding to the temperatures where the compensation is most effective.

In addition to the high Nernst thermopower, topological semimetals usually have low resistivities. Notably, although WTe$_2$ shows giant magnetoresistance, its resistivity (at 9 T) has the same order of magnitude as general thermoelectric semiconductors such as (Bi,Sb)$_2$Te$_3$ (at 0 T) (Supplementary Fig. 5a)[38]. Consequently, the Nernst power factor, $PF = S_{yx}^2\rho_{yy}$, of WTe$_2$ (Fig. 4b) reaches an ultrahigh value of 3 WmK$^{-2}$ (at ~15 K), which is 2–3 orders higher than the general Seebeck power factors. The maximum Nernst thermoelectric figure of merit, $zT = (PF/\kappa)T$, where $\kappa$ is the thermal conductivity and $T$ is the absolute temperature, reaches around 0.2 at 10 K (inset in Fig. 4b). This is highly promising, as it is of the same order of magnitude as those of classic longitudinal thermoelectric semiconductors (Supplementary Fig. 5b, c). Herein $\rho_{xx}$ is used for calculating the Nernst $PF$ of WTe$_2$ instead of $\rho_{yy}$, as the WTe$_2$ single crystal is too short

to measure its resistivity along the $b$-axis. Nevertheless, we can safely assume nearly isotropic bonding within the layers, indicated by the weak anisotropy of both electron and hole pockets for WTe$_2$[23] and hence, a very small difference between $\rho_{xx}$ and $\rho_{yy}$. Reasonably, the Nernst $PF$ of WTe$_2$ is several orders of magnitude higher than the Seebeck $PF$ values, including those of the best thermoelectrics below room temperature such as Bi$_{1-x}$Sb$_x$[39] and (Bi,Sb)$_2$Te$_3$[38]. This is remarkable as few materials have been found with such high $PF$ values to date.

Note that the longitudinal thermoelectric materials based on the Seebeck effect have few representatives below 300 K. Bi$_{1-x}$Sb$_x$ is the best thermoelectric material below 300 K, showing the best performance at ~100–150 K. Rare materials have been reported to present high thermoelectric performance at extremely low temperatures. In this regard, the high transverse thermoelectric performance of WTe$_2$ is of significant interest for solid-state cooling at low temperatures, especially for niche applications where vibration-free operation is more important than the cost of

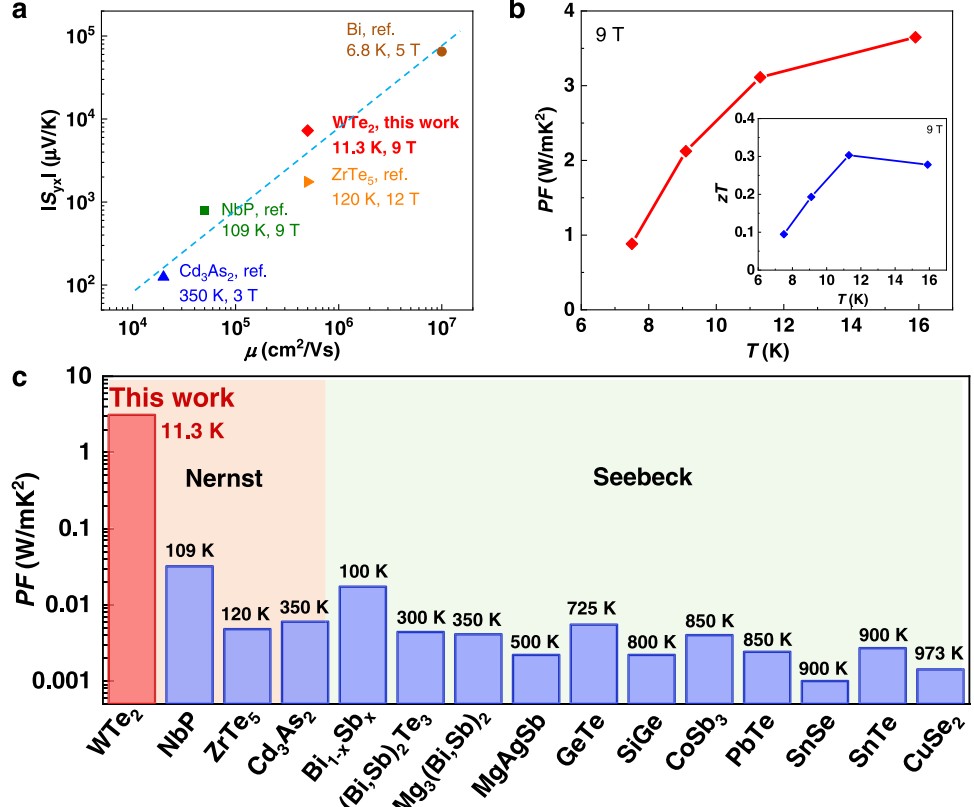

**Fig. 4 Comparison of high-performance thermoelectric semiconductors. a** Nernst thermopower and mobility of various topological semimetals and Bi. **b** temperature dependence of Nernst power factor of WTe$_2$, inset shows the $zT$ values. **c** Nernst power factor of various topological semimetals including NbP[8,40], ZrTe$_5$[36] and Cd$_3$As$_2$[37], and a comparison with the Seebeck power factors of high-performance thermoelectric semiconductors including Bi$_{1-x}$Sb$_x$[39], (Bi,Sb)$_2$Te$_3$[38], Mg$_3$(Bi,Sb)$_2$[41], MgAgSb[42], GeTe[43], SiGe[44], CoSb$_3$[45], PbTe[46], SnSe[47], SnTe[48], and CuSe$_2$[49].

providing the magnetic field. Most importantly, it demonstrates the great potential of van der Waals bonded topological materials as a platform for developing flexible thermoelectrics.

Fig. 4c summarizes the Nernst *PF* values of WTe$_2$ and other topological semimetals, including NbP[8,40], ZrTe$_5$[36] and Cd$_3$As$_2$[37], with a comparison of the Seebeck *PF* values of most of the well-known longitudinal thermoelectric semiconductors[41–49]. The topological semimetals, particularly WTe$_2$, have high *PF* values, which are even 2–3 orders of magnitude higher than those of the best thermoelectric semiconductors. For practical applications, both high *PF* and high *zT* are desirable. Since the thermal conductivities of WTe$_2$ single crystals (see Supplementary Fig. 6) are significantly higher than those of general thermoelectric materials, there is a large room to decrease the thermal conductivity of WTe$_2$. Therefore, strategies like alloying to reduce the thermal conductivity could be effective for further enhancing its thermoelectric performance.

## Discussion
In summary, the transformative WTe$_2$ Weyl semimetal shows a high Nernst power factor of ~3 Wm$^{-1}$K$^{-2}$ and a promising Nernst *zT* of 0.3 at 11.3 K at 9 T. The Nernst signal shows a linear dependence on the magnetic field, with the maximum Nernst signal reaching 7000 μVK$^{-1}$ at 11.3 K and 9 T. The Ettingshausen signal reaches 5 × 10$^{-5}$ KA$^{-1}$m at 23.1 K and 9 T. These large transverse thermoelectric signals are attributed to the extremely high mobility, low-effective masses of the Fermi pockets, and nearly perfect compensation of electrons and holes. Compared with conventional longitudinal thermoelectric materials, which usually perform best above room temperature, WTe$_2$

demonstrates the great potential of topological semimetals for thermoelectric applications at extremely low temperatures.

In future studies, other van der Waals bonded topological semimetals with linear sharp-band dispersion characteristics should be investigated as they could be excellent materials to achieve high transverse thermoelectric performance. Searching for materials with high mobility and a compensation temperature near room temperature would be of great interest for thermoelectric cooling at 300 K. Novel materials with high thermoelectric performance requiring small external magnetic fields are of great significance for practical applications. To efficiently enhance the transverse thermoelectric figure of merit for WTe$_2$, there are many conceivable options for furthering reducing its thermal conductivity, such as chemical substitution or microstructural engineering. Moreover, it would be interesting to measure the thermoelectric transport properties of twisted or shaped WTe$_2$ to investigate its suitability for flexible thermoelectrics. In short, with the achievement of high thermoelectric performance, we expect the application of topological materials for solid-state cooling under challenging conditions, such as low-dimensional devices, flexible devices, and ultralow temperatures.

## Methods
**Sample preparation**. WTe$_2$ single crystals were grown using a self-flux method with an elemental ratio of W:Te = 1:80. W powder (Alfa Aesar, 99.99%) and Te chunks (Alfa Aesar, 99.999%) were mixed and then placed in an alumina crucible. Subsequently, they were sealed in a quartz tube under partial argon pressure. The sealed tube was then heated to 1100 °C in 3 days, kept at this temperature for 48 h, and then slowly cooled to 500 °C at a rate of 2 °C/h. Finally, the single crystals were separated from the flux by centrifugation. Ribbon-shaped layered crystals with shiny surfaces were obtained.

## Measurements of the transport properties

**Measurements of the transport properties.** The transport properties were measured in the *ab*-plane direction: a current or temperature gradient was applied along the *a*-axis (100), and the Nernst and Ettingshausen signals were measured along the *b*-axis (010) with a magnetic field applied along *c*-axis (001) direction, perpendicular to the layers. The longitudinal resistivities ($\rho_{xx}$) and Hall resistivities ($\rho_{yx}$) were measured using a physical property measurement system (PPMS9, Quantum Design) in the electrical-transport mode with a standard four-probe method. The Nernst thermopower, Ettingshausen temperature gradient, Seebeck coefficient, and thermal conductivity were measured under high vacuum in the PPMS by a standard four-contact, steady-state method (Supplementary Fig. 7). The Nernst thermopower, Seebeck coefficient, and thermal conductivity were measured in sweep-field mode, while the Ettingshausen effect was measured in set-field mode. All measured data were field-symmetrized and antisymmetrized to correct for contact misalignments. Additional transport characterization under magnetic fields up to 65 T were carried out in a 70 T pulse magnet using a $^3$He insert.

## Data availability

All the data supporting the plots within this paper and the findings of this study are available from the corresponding author upon request.

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

## Acknowledgements

This work was financially supported by the European Research Council (ERC Advanced Grant No. 742068 'TOPMAT'); the DFG through the SFB 1143 project (ID. 247310070) and the Würzburg-Dresden Cluster of Excellence on Complexity and Topology in Quantum Matter ct.qmat (EXC2147, project ID. 390858490). Y.P. acknowledges the financial support from the Alexander von Humboldt Foundation. We acknowledge the support of the HLD at HZDR, member of the European Magnetic Field Laboratory.

## Author contributions

Y.P. designed the experiments, grew the single crystal, conducted the crystallinity and composition characterization, and measured the transport properties. B.H. programmed the LabView sequence for the thermal transport properties measurements. T.H. supported the high-field transport measurements. D.C. helped analyze the oscillation data. W.S. helped the measurements of the transport properties. C.F. supervised the project. All authors discussed the results and contributed to the preparation of the manuscript.

## Funding

## Competing interests

The authors declare no competing interests.
