## [Peer Review File · Nature Communications]

Ultrahigh transverse thermoelectric power factor in flexible Weyl semimetal WTe₂REVIEWER COMMENTS

Reviewer #1 (Remarks to the Author):

The authors report ultrahigh transverse thermoelectric power factor in Weyl semimetal WTe₂, with the Nernst power factor reaching 3 W/mK² and the Ettingshausen signal being about $2 \times 10^{-3} \text{ K A}^{-1}$. Such a feature is attributed to the extremely high carrier mobility, low effective mass of the Fermi pockets, and the nearly perfect compensation of electron and hole carriers. The high transverse thermoelectric power factor, in conjunction of the mechanical flexibility of WTe₂, renders this material a great candidate for small thermoelectric device, compared to the traditional devices based on longitudinal Seebeck effect. This finding, particularly considering its great potential application, warrants the publication of this work in Nature Communications.

There are a couple of minor issues to be addressed before the manuscript is published.

1) On Page 3, the authors mentioned that the studies on the transverse thermoelectric effect of WTe₂ single crystals is rare. Surprisingly, they did not mention their early work (Nano. Lett. 18, 6591 (2018)) at all which reports unconventional Nernst effect of WTe₂ single crystal flakes.

2) On Page 6, the authors need to move the definition of Eq. (4) upfront, since the same parameters are already used in Eq. (1).

Reviewer #2 (Remarks to the Author):

This paper reports on systematic measurements of transverse thermoelectric properties in the Weyl semimetal WTe₂. However, the observation of the large Nernst effect in WTe₂ was reported in 2015 by Zhu et al. (Ref. 21). Although the observed Nernst thermopower is extremely high, its practical application seems to be impossible because it works only at very low temperatures and requires a huge magnetic field. Nevertheless, the systematic experiments and detailed analyses reported in this paper are worth publishing and useful for fundamental condensed matter physics. Thus, I recommend the transfer of this paper to Communications Physics after addressing the following issues.

1) In the Ettingshausen experiments, the authors show the transverse temperature gradient normalized by charge current density. Using this data, the authors should estimate the Ettingshausen coefficient and confirm the Onsager reciprocal relation between the Nernst and Ettingshausen effects quantitatively.

2) Comment about the sentence "The relatively large current induced Joule heating results in a large temperature drift and prevented us from reaching below 20 K" on page 5: In this condition, how did the authors measure the difference between the actual sample temperature and sample stage temperature? The transverse axis of Fig. 2e should be the actual sample temperature, not the stage temperature.

3) The performance of the transverse thermoelectric conversion is evaluated by the adiabatic or isothermal figures of merit depending on the thermal boundary conditions. Which definition was used for estimating the ZT value in Fig. S5? This point should be carefully taken into account when the Nernst performance is compared with the Seebeck performance.

4) The Ettingshausen cooler was investigated actively in 1960s (e.g., APL 2, 145 (1963) and APL 4, 77 (1964)). However, the practical application was not realized because the operation of the Ettingshausen cooler requires a huge magnetic field. This point should be mentioned.

5) Comment about the sentence "Although liquid helium enables cooling 300 down to ~4 K it is very expensive. A Nernst Ettingshausen cooler can therefore be an excellent alternative" on page 9:

To drive the large Ettingshausen effect in WTe₂ at high magnetic fields and low temperatures, liquid helium and/or high-power refrigerator are necessary. Thus, I do not think that the large Ettingshausen effect in WTe₂ is useful in this context. To confirm the usefulness, the coefficient of performance should be compared between the WTe₂-based Ettingshausen cooler and conventional technologies.

6) In Fig. 1e,f, the directions of the a, b, and c axes should be depicted to clearly show the experimental setup.

Reviewer #3 (Remarks to the Author):

In this manuscript Pan et al study the thermoelectric properties of bulk WTe₂, focusing specifically on the Nernst and Ettingshausen effects. They find a very large Nernst effect, owing presumably to the high mobility and near-complete compensation of carriers. The results are impressive, and could be useful for future Nernst-based thermoelectric devices. I recommend this paper for publication after the authors have considered the comments and suggestions below.

1. The discussion of the band structure of bulk WTe₂ is sometimes confusing. For example, I don't see how Fig. 1(c) can be a representation of a type-II Weyl semimetal, which is gapless. This figure also gives the impression of small electron and hole pockets, while a type-II Weyl semimetal would have large pockets.

2. A related question is whether the carrier concentrations inferred from fitting the Hall conductivity ($\sim 10^{19} \text{ cm}^{-3}$) are consistent with what is known from the WTe₂ band structure. Is this what one would have expected from the known band structure, given a Fermi level at the Weyl point?

The reported frequencies from SdH measurements should give another estimate of the carrier concentration. A rough estimate assuming a spherical Fermi surface seems (to me) to give a consistent number for the carrier concentration, $\sim 10^{19}$. But it would be nice if the authors can check this and make a comment.

3. There seem to be some inconsistencies in the notation. For example, in equation (1) the Seebeck coefficient is α , but elsewhere it is S_{xx} . Also, in equation (3) $|\alpha|$ is used to denote the absolute value of the Seebeck coefficient -- is it also used to mean the absolute value in equation (1)? This seems particularly important because α enters equation (1) as $\alpha_e - \alpha_h$, which is either large or small depending on whether $|\alpha|$ is supposed to mean the absolute value.

4. The peak in S_{xy} at $T \sim 10 \text{ K}$ does not seem to have an especially clear interpretation in the text. The authors write that "most likely, this is the temperature where the electrons and holes compensate each other most efficiently." Does this mean that the Fermi energy is ~ 10

K \sim 1 meV below the Weyl point?

An alternative possibility is that somewhere around 10 K there is a sharp increase in phonon scattering, which lowers the mobility. Such an interpretation seems perhaps consistent with the behavior of the thermal conductivity shown in the supplemental figure S6. Can the authors comment on this possibility?

5. The change of sign in S_{xx} with T also seems not especially clear. Certainly one can expect that there are more thermally-excited electrons when T becomes comparable to the Fermi energy, but there are also more holes, so the sign of S_{xx} wouldn't obviously invert. Is there some reason why the electrons have higher mobility than the holes?

Of course, I understand that these may be tricky questions, and it's okay with me if there is no clear answer to them at the moment. But if the authors have some additional understanding or intuition then it would be valuable to add to the text.

COMMENTS TO AUTHOR:

Reviewer #1: The authors report ultrahigh transverse thermoelectric power factor in Weyl semimetal WTe₂, with the Nernst power factor reaching 3 W/mK² and the Ettingshausen signal being about 2 × 10⁻³ K A⁻¹. Such a feature is attributed to the extremely high carrier mobility, low effective mass of the Fermi pockets, and the nearly perfect compensation of electron and hole carriers. The high transverse thermoelectric power factor, in conjunction of the mechanical flexibility of WTe₂, renders this material a great candidate for small thermoelectric device, compared to the traditional devices based on longitudinal Seebeck effect. This finding, particularly considering its great potential application, warrants the publication of this work in Nature Communications. There are a couple of minor issues to be addressed before the manuscript is published.

Response: We greatly appreciate the reviewer's approval of this work. We have addressed the minor issues point-by-point as shown below.

Specific comments:

(1) On Page 3, the authors mentioned that the studies on the transverse thermoelectric effect of WTe₂ single crystals is rare. Surprisingly, they did not mention their early work (Nano. Lett. 18, 6591 (2018)) at all which reports unconventional Nernst effect of WTe₂ single crystal flakes.

Response to comment (1):

Thanks a lot for your kind remind. Previously we focused on the bulk crystals, in the revised manuscript we have added our early work (Nano. Lett. 18, 6591 (2018)) in flakes as ref. 24.

“...studies on the transverse thermoelectric performance of WTe₂ single crystals from the point-of-view of thermoelectric applications are rare.^{23,24}” (main text, page 5)

24. Rana, K. G. et al. Thermopower and unconventional Nernst effect in the predicted Type-II Weyl semimetal WTe₂. *Nano Lett.* **18**, 6591–6596 (2018). [10.1021/acs.nanolett.8b03212](https://doi.org/10.1021/acs.nanolett.8b03212) (references, page 20)

(2) On Page 6, the authors need to move the definition of Eq. (4) upfront, since the same parameters are already used in Eq. (1).

Response to comment (2):

Thanks for the careful suggestion. We have moved the definition of various parameters upfront when they used the first time in Eq. (1). Detailed revisions are shown below.

$$N = \frac{(N_e \sigma_e + N_h \sigma_h)(\sigma_e + \sigma_h) + (N_e \sigma_e \mu_h - N_h \sigma_h \mu_e)(\sigma_e \mu_h - \sigma_h \mu_e) B^2 + \sigma_e \sigma_h (\mu_h + \mu_e)(\alpha_h - \alpha_e)}{(\sigma_e + \sigma_h)^2 + (\sigma_e \mu_h - \sigma_h \mu_e)^2 B^2}, \quad (1)$$

where N_e and N_h , σ_e and σ_h , μ_e and μ_h , α_e and α_h denote the Nernst coefficient, conductivity, mobility, and Seebeck coefficient of electrons and holes, respectively, and B is the magnetic field.²³ (page 7)

$$\sigma_{xy} = \frac{\rho_{yx}}{\rho_{xx}^2 + \rho_{yx}^2} = \left[\frac{-n_e \mu_e^2}{1 + (\mu_e B)^2} + \frac{n_h \mu_h^2}{1 + (\mu_h B)^2} \right] eB \quad (4)$$

where ρ_{yx} and ρ_{xx} are the Hall and longitudinal resistivity (ρ_{xx} and ρ_{yx} are shown in **Fig. 3(a)** and **(b)**), n_e , and n_h are the charge carrier concentration of electrons and holes, respectively.” (page 10)

Reviewer #2: This paper reports on systematic measurements of transverse thermoelectric properties in the Weyl semimetal WTe2. However, the observation of the large Nernst effect in WTe2 was reported in 2015 by Zhu et al. (Ref. 21). Although the observed Nernst thermopower is extremely high, its practical application seems to be impossible because it works only at very low temperatures and requires a huge magnetic field. Nevertheless, the systematic experiments and detailed analyses reported in this paper are worth publishing and useful for fundamental condensed matter physics. Thus, I recommend the transfer of this paper to Communications Physics after addressing the following issues.

Response: We are glad that the reviewer thinks this work is systematic and useful for fundamental condensed matter physics. While for the concerns about the novelty of this work and the potential applications, we believe the following clarifications would be helpful.

We noticed that in 2015 Zhu et al. reported the Nernst effect in WTe2 (cited as ref. 23), however, the motivation as well as the findings of present work is very different from their report. **First**, we report the Nernst thermopower in a wide temperature range from a point of view of thermoelectric performance, pointing out the maximum Nernst thermopower shows at ~10 K, while Zhu et al. focused on resolving the Fermi surfaces of WTe2 based on the oscillation at extremely low temperatures below 6 K. **Second**, the residual resistivity ratio (RRR) in the present work is much higher than previous reports, including Zhu’s work (ref. 23), indicating better quality as well as the better performance of the single crystals in the present work. **Third**, we focus on the transverse thermoelectric performance including power factor and thermoelectric figure of merit in WTe2, which to our best knowledge, have never been reported by others before. **Last but not least**, beyond Nernst effect, we also report Ettingshausen effect in WTe2, which to our best knowledge, have never been reported by others before. Therefore, we believe the findings in the present work are new and worth the publication in high quality journals.

We also understand that the practical applications of WTe2 at present is very challenging, however, it is a significant pioneer example demonstrating the great possibility of topological materials like WTe2 for future flexible thermoelectric applications. We have carefully considered the valuable comments and suggestions, based on which we have appropriately revised the manuscript. Detailed revisions can be found in the response to comment (4) and (5).

Followings are the point-by-point response, we hope it addresses your concerns.

Specific comments:

(1) In the Ettingshausen experiments, the authors show the transverse temperature gradient normalized by charge current density. Using this data, the authors should estimate the Ettingshausen coefficient and confirm the Onsager reciprocal relation between the Nernst and Ettingshausen effects quantitatively.

Response to comment (1):

Thanks for the nice comment. According to the Onsager reciprocal relation, the Ettingshausen coefficient P can be determined by the Nernst coefficient N (isothermal) via the Bridgman relation:

$$P_{xy} = N_{yx}T/\kappa_{xx}$$

In which κ is the thermal conductivity and T is the absolute temperature. Noteworthy, the measured Nernst thermopower in the present work is the adiabatic Nernst instead of the isothermal Nernst (i.e. $\nabla_y T = 0$ is not guaranteed). Moreover, in our measurement set up, due to the sample geometry, both the Ettingshausen and the Nernst effects are measured in the transverse direction (along b-axis) by applying the current/heat flux along the longitudinal direction (along a-axis), i.e. we measured both P_{yx} and N_{yx} . Additionally, note that $P_{xy} \neq P_{yx}$ in WTe2 due to its point group of $mm2$ (Y. C. Akgozt and G. A. Saunders, J. Phys. C: Solid State Phys. 8, 2962, 1975). In this case, the experimental data (P_{yx} and N_{yx}) do not have to obey the Onsager reciprocal relation.

Though the measured results cannot be related via the Onsager reciprocal relation, we believe it is a very good point to make a clarification. Therefore, we have estimated P_{xy} from N_{yx} . Taking the data at 20.7 K as an example, $P_{xy} = N_{yx}T/\kappa_{xx} = 2 \times 10^{-5} \text{ KA}^{-1}\text{m}$, which is close to the measured P_{yx} at 23.1 K ($P_{yx} = (\nabla T_{yx}/j_{xx})/B_z = 0.6 \times 10^{-5} \text{ KA}^{-1}\text{m}$). We have added the above discussion in the revised manuscript and supplementary, as highlighted on page 5 and 2, respectively, which are also shown below.

“In addition, a rough estimation of the Ettingshausen coefficient can be obtained from the Nernst coefficient based on the Onsager relation, which can be found in the Supplementary Information.” (main text, page 8)

“According to the Onsager reciprocal relation, the Ettingshausen coefficient P can be determined by the isothermal Nernst coefficient N via the Bridgman relation:

$$P_{xy} = N_{yx}T/\kappa_{xx}$$

In which κ is the thermal conductivity and T is the absolute temperature. Herein, the reported Nernst and Ettingshausen performances do not have to obey the above relation because: (a) the

Nernst effect is measured under adiabatic condition instead of isothermal condition; (b) both the Ettingshausen and the Nernst effects are measured in the transverse direction (along b-axis) by applying the current/heat flux along the longitudinal direction (along a-axis), i.e. P_{yx} and N_{yx} (note that $P_{xy} \neq P_{yx}$ in WTe_2 due to its point group of $mm2$). Nevertheless, we roughly estimated P_{xy} using the adiabatic N_{yx} . Taking the data at 20.7 K as an example, $P_{xy} = N_{yx}T/\kappa_{xx} = 2 \times 10^{-5} \text{ KA}^{-1}\text{m}$, which is close to the measured P_{yx} at 23.1 K ($P_{yx} = (\nabla T_{yx}/j_{xx})/B_z = 0.6 \times 10^{-5} \text{ KA}^{-1}\text{m}$).” (supplementary, page 3)

(2) Comment about the sentence "The relatively large current induced Joule heating results in a large temperature drift and prevented us from reaching below 20 K" on page 5: In this condition, how did the authors measure the difference between the actual sample temperature and sample stage temperature? The transverse axis of Fig. 2e should be the actual sample temperature, not the state temperature.

Response to comment (2):

Thanks for the careful comment.

The actual sample temperature can be determined by analyzing the hot side and cold side temperature. For example, if the state temperature is 20 K, we first assume that the actual temperature is 20 K, then we know the increased temperature on both hot side and cold side via $V_H/V_{20, TC}$ and $V_C/V_{20, TC}$, respectively. Here V_H and V_C is the measured voltage at the hot side and cold side, $V_{20, TC}$ is the known thermocouple voltage (per Kelvin) at 20 K. This indicate that the sample temperature has been increased by $(V_H/V_{20, TC} + V_C/V_{20, TC})/2$, which we define as δ . Then we assume the “actual” sample temperature is $(20+\delta)$ K. When $(V_H/V_{20+\delta} + V_C/V_{20+\delta})/2 = \delta$, we determine δ and the actual sample temperature. Usually the δ is relatively small, and by trying 2-3 different δ values, we can finally determine the actual sample temperature.

We’ve changed the actual sample temperature instead of the state temperature in Fig. 2, along with the necessary revision of the data owing to the varied temperatures (with a higher actual sample temperature, we should use a larger relative voltage, which gives to a smaller actual temperature gradient and thus a larger Nernst signal). The revised Fig. 2 are shown below.

Figure 2. Nernst, Ettingshausen, and Seebeck effect in WTe₂. **a** Nernst signal from 7.5 K to 30.3 K. **b** Ettingshausen signal from 23.1 K to 103.6 K. **c** Seebeck coefficient from 7.5 K to 100.9 K. **d** Nernst signal as a function of temperature. **e** Ettingshausen signal as a function of temperature. **f** Temperature dependent Seebeck coefficient at 0 T of a WTe₂ single crystal. Both the Nernst and Ettingshausen effects show large, non-saturating signals up to 9 T. The maximum Nernst effect is observed in the range of 10–15 K, and the maximum Ettingshausen effect observed in the present work was at 23.1 K, which is the lower temperature limit of our measurements. (figure and figure caption, page 8-9)

(3) The performance of the transverse thermoelectric conversion is evaluated by the adiabatic or isothermal figures of merit depending on the thermal boundary conditions. Which definition was used for estimating the ZT value in Fig. S5? This point should be carefully taken into account when the Nernst performance is compared with the Seebeck performance.

Response to comment (3):

Thanks for the careful suggestion. The transverse thermoelectric figure of merit is evaluated by using the adiabatic Nernst thermopower, adiabatic thermal conductivity, and adiabatic resistivity. We've added the definition of adiabatic condition in Fig. S5, which is also shown below.

“Herein, to calculate the Nernst zT of WTe₂, adiabatic Nernst thermopower, adiabatic thermal conductivity, and adiabatic resistivity are used.” (supplementary, page 4)

(4) The Ettingshausen cooler was investigated actively in 1960s (e.g., APL 2, 145 (1963) and APL 4, 77 (1964)). However, the practical application was not realized because the operation of the Ettingshausen cooler requires a huge magnetic field. This point should be mentioned.

Response to comment (4):

We thank you very much for the good point. We agree that the requirement of large external magnetic field would limit the applications of the Nernst thermoelectric devices. As a consequence, a great breakthrough of the Nernst performance, particularly under a relatively low magnetic field (the same for anomalous Nernst effect), would be of great significance for its potential applications, probably in niche applications where the cost of the magnetic field is less important than the solid-state cooling without vibration, and of course, the use of permanent magnet should be taken into consideration. We've added more discussions on this point, as shown on page 6 and 16, which are also shown below. "Ettingshausen coolers have been investigated actively in the 1960s,^{26,27} but the requirement of a large external magnetic field limits their practical application. Therefore, a great breakthrough of the Nernst–Ettingshausen performance particularly under a relatively small magnetic field, is critical for future niche applications, e.g., where vibration-free operation is more important than the cost of providing a strong magnetic field. The advantages of permanent magnets should also be considered." (main text, page 6)

"Novel materials with high thermoelectric performance requiring small external magnetic fields are of great significance for practical applications." (main text, page 16)

26. Cuff, K. F. et al. The thermomagnetic figure of merit and Ettingshausen cooling in Bi-Sb alloys. *Appl. Phys. Lett.* **2**, 145–146 (1963). [10.1063/1.1753817](https://doi.org/10.1063/1.1753817) (references, page 20)

27. Harman, T. C., Honig, J. M., Fischler, S., Paladino, A. E. & Button, M. J. Oriented single-crystal bismuth Nernst-Ettingshausen refrigerators. *Appl. Phys. Lett.* **4**, 77–79 (1964). [10.1063/1.1753970](https://doi.org/10.1063/1.1753970) (references, page 20)

(5) Comment about the sentence "Although liquid helium enables cooling 300 down to ~4 K it is very expensive. A Nernst Ettingshausen cooler can therefore be an excellent alternative" on page 9: To drive the large Ettingshausen effect in WTe₂ at high magnetic fields and low temperatures, liquid helium and/or high-power refrigerator are necessary. Thus, I do not think that the large Ettingshausen effect in WTe₂ is useful in this context. To confirm the usefulness, the coefficient of performance should be compared between the WTe₂-based Ettingshausen cooler and conventional technologies.

Response to comment (5):

We thank you for your comment from the practical application viewpoint. We've deleted the sentence since we agree that WTe₂ at present cannot compete with the commercial helium cooling. Nevertheless, we would like to highlight that the present work, realizing a high thermoelectric performance and mechanical flexibility, can be instructive for looking for more fantastic topological materials for future applications with the concept of flexible transverse thermoelectrics. We've made appropriate changes in the revised manuscript, as shown on page 1, 3, 14 and below.

"In this work, the combination of the exceptional Nernst–Ettingshausen performance and excellent mechanical transformative ability of WTe₂ would be instructive for flexible micro-/nano-thermoelectric devices." (page 1)

“The present work highlights the role of flexible topological semimetals for thermoelectric cooling, particularly as flexible devices, in niche applications.” (page 3)

“Rare materials have been reported to present high thermoelectric performance at extremely low temperatures. In this regard, the high transverse thermoelectric performance of WTe_2 is of significant interest for solid-state cooling at low temperatures, especially for niche applications where vibration-free operation is more important than the cost of providing the magnetic field. Most importantly, it demonstrates the great potential of van der Waals bonded topological materials as a platform for developing flexible thermoelectrics.” (page 14)

(6) In Fig. 1e,f, the directions of the a, b, and c axes should be depicted to clearly show the experimental setup.

Response to comment (6):

We thank you for your suggestion. We’ve revised Fig. 1 e,f by including the directions of a, b, and c axes in the experimental setup, as shown on page 5 in the revised manuscript and below.

Figure 1. WTe_2 basic properties and the Nernst–Ettingshausen effect. ... e Schematic illustrations of Nernst effect. **f** Ettingshausen effect. **The orientation of the experimental setup corresponding to the crystal axes is shown.** Both the Nernst and Ettingshausen devices need only one material, and therefore have lower complexity than Seebeck devices. (figure and figure caption, page 5)

Reviewer #3: In this manuscript Pan et al study the thermoelectric properties of bulk WTe_2 , focusing specifically on the Nernst and Ettingshausen effects. They find a very

large Nernst effect, owing presumably to the high mobility and near-complete compensation of carriers. The results are impressive, and could be useful for future Nernst-based thermoelectric devices. I recommend this paper for publication after the authors have considered the comments and suggestions below. Of course, I understand that these may be tricky questions, and it's okay with me if there is no clear answer to them at the moment. But if the authors have some additional understanding or intuition then it would be valuable to add to the text.

Response: We greatly appreciate the reviewer for approving that this work is impressive and could be useful for future thermoelectric applications. We've considered your comments and suggestions carefully and made appropriate changes/discussions in the revised manuscript. Please see the responses below.

Specific comments:

(1) The discussion of the band structure of bulk WTe₂ is sometimes confusing. For example, I don't see how Fig. 1(c) can be a representation of a type-II Weyl semimetal, which is gapless. This figure also gives the impression of small electron and hole pockets, while a type-II Weyl semimetal would have large pockets.

Response to comment (1):

Thank you for pointing this out. We've changed Fig. 1(c) as well as the corresponding text in the revised manuscript on page 5, as shown below. The schematic band structure of WTe₂ presents the compensation behavior of electrons and holes at the Fermi energy. Moreover, according to previous studies, the Weyl points are ~50 meV above the Fermi energy if there is a perfect compensation of electrons and holes (A. A. Soluyanov et al. Nature 527, 495-498, 2015; S. Sykora et al. Phy. Rev. Research 2, 033041, 2020). Since the Weyl points of WTe₂ are not along the high symmetry points, no gap is illustrated (A. A. Soluyanov et al. Nature 527, 495-498, 2015; S. Sykora et al. Phy. Rev. Research 2, 033041, 2020). We've also revised the corresponding text and added the literatures as ref. 21 and 22.

... Fig. 1(c) shows the compensation behavior of electrons and hole at the Fermi energy, which leads to extremely large magnetoresistance; and the Weyl points are ~50 meV above the Fermi energy.^{21,22}(main text, page 4)

Figure 1. WTe₂ basic properties and the Nernst–Ettingshausen effect. ... c Schematic of the band structure of WTe₂ Weyl semimetal. **Electrons and holes show nearly perfect compensation at the Fermi energy. The green dots show a pair of Weyl points, which are above the Fermi energy.** (figure and figure caption, page 5)

21. Soluyanov, A. A. et al. Type-II Weyl semimetals. *Nature* **527**, 495–498 (2015). [10.1038/nature15768](https://doi.org/10.1038/nature15768) (References, page 20)

22. Sykora, S. et al. Disorder-induced coupling of Weyl nodes in WTe₂. *Phys. Rev. Res.* **2**, 033041 (2020). [10.1103/PhysRevResearch.2.033041](https://doi.org/10.1103/PhysRevResearch.2.033041) (References, page 20)

(2) A related question is whether the carrier concentrations inferred from fitting the Hall conductivity ($\sim 10^{19} \text{ cm}^{-3}$) are consistent with what is known from the WTe₂ band structure. Is this what one would have expected from the known band structure, given a Fermi level at the Weyl point? The reported frequencies from SdH measurements should give another estimate of the carrier concentration. A rough estimate assuming a spherical Fermi surface seems (to me) to give a consistent number for the carrier concentration, $\sim 10^{19}$. But it would be nice if the authors can check this and make a comment.

Response to comment (2):

Thanks for the good comment. We have roughly estimated the carrier concentration from the SdH oscillation assuming a spherical Fermi surface, $n = (1/3\pi^2)(2eF/\hbar)^{3/2}$, in which F is the frequency. Table S1 shown below illustrates the results. The results show that for the total hole concentration, $n_h = 2*(n_{h1} + n_{h2}) = 3.28E19 \text{ cm}^{-3}$, for the total electron concentration, $n_e = 2*(n_{e1} + n_{e2}) = 3.64E19 \text{ cm}^{-3}$ (the carrier concentration needs to be doubled owing to two sets of pockets in the Brillouin zone). The SdH estimated carrier concentration is close to the Hall results, within the $\sim 10^{19} \text{ cm}^{-3}$ order. Small differences can be due to the anisotropy of the Fermi surface.

Zhu et al. (PRL 114, 176601, 2015) have mapped the Fermi surface of WTe₂, and reported a compensated carrier concentration of $\sim 10^{19} \text{ cm}^{-3}$ as well. However, because

of the compensation proof, the Fermi level is not at the Weyl point (A. A. Soluyanov et al. Nature 527, 495-498, 2015).

We have included the discussion in the revised manuscript (page 10) and supplementary information (page 4), as shown below.

“The electron and hole concentrations are very similar, with an order of 10^{19} cm^{-3} ; this was also confirmed by the estimated values from the oscillation (Supplemental Table 1).”(page 10)

“Assuming a spherical Fermi surface, we can roughly estimate the carrier concentration n from the SdH oscillation, $n = (1/3\pi^2)(2eF/\hbar)^{3/2}$, in which F is the frequency, e is the elementary charge, and \hbar is the reduced Planck constant. **Table 1** illustrates the results. For the total hole concentration, $n_h = 2 \times (n_{h1} + n_{h2}) = 3.28\text{E}19 \text{ cm}^{-3}$, for the total electron concentration, $n_e = 2 \times (n_{e1} + n_{e2}) = 3.64\text{E}19 \text{ cm}^{-3}$, considering that there are two sets of pockets in the Brillouin zone. The SdH estimated carrier concentration is close to the Hall results, within the $\sim 10^{19} \text{ cm}^{-3}$ order. Small differences can be due to the anisotropy of the Fermi surface.” (supplementary, page 4)

Table 1 Estimated carrier concentration from SdH oscillations.

Pockets	Frequency (T)	Carrier concentration (cm^{-3})
hole 1	94	0.51E19
electron 1	130	0.84E19
electron 2	144	0.98E19
hole 2	159	1.13E19

(3) There seem to be some inconsistencies in the notation. For example, in equation (1) the Seebeck coefficient is α , but elsewhere it is S_{xx} . Also, in equation (3) α is used to denote the absolute value of the Seebeck coefficient -- is it also used to mean the absolute value in equation (1)? This seems particularly important because α enters equation (1) as $\alpha_e - \alpha_h$, which is either large or small depending on whether α is supposed to mean the absolute value.

Response to comment (3):

Thanks for the nice suggestions. We’ve re-organized the parameters used in the equations. In the revised manuscript, only α denotes the Seebeck coefficient. We’ve revised it in both equation (1) and (3). In addition, please note that in equation (1) it is the Seebeck coefficient, instead of the absolute value of Seebeck coefficient.

$$\alpha = (\alpha_e \sigma_e + \alpha_h \sigma_h) / (\sigma_e + \sigma_h), \quad (3)$$

(4) The peak in S_{xy} at $T \sim 10 \text{ K}$ does not seem to have an especially clear interpretation in the text. The authors write that "most likely, this is the temperature where the electrons and holes compensate each other most efficiently." Does this mean that the Fermi energy is $\sim 10 \text{ K} \sim 1 \text{ meV}$ below the Weyl point? An alternative possibility is that somewhere around 10 K there is a sharp increase in phonon scattering, which lowers the mobility. Such an interpretation seems perhaps consistent with the behavior of the thermal conductivity shown in the supplemental figure S6. Can the authors comment on this possibility?

Response to comment (4):

Thanks for the instructive suggestions. As for the Fermi energy, we noticed that it would be ~ 50 meV below the Weyl point when the electrons and holes are perfectly compensated, according to the literature (A. A. Soluyanov et al. Nature 527, 495-498, 2015). In this case, we agree that the sharp increase of Seebeck coefficient at ~ 10 K is probably due to the phonon drag effect, particularly considering that there is a sign change of the Seebeck coefficient. And it is true that the sharp turning point in Seebeck coefficient happens at a temperature lower than the temperature where the peak value of thermal conductivity is observed, which again indicates the possibility of phonon drag effect. We've included the possibility of phonon drag effect on the sharp increase in Seebeck coefficient, as shown on page 9 and also below.

“The turning point can be understood considering two aspects. First, phonon scattering sharply increases above 10 K, leading to the phonon drag effect on the turning point of the Seebeck coefficient.” (page 9)

(5) The change of sign in S_{xx} with T also seems not especially clear. Certainly one can expect that there are more thermally-excited electrons when T becomes comparable to the Fermi energy, but there are also more holes, so the sign of S_{xx} wouldn't obviously invert. Is there some reason why the electrons have higher mobility than the holes?

Response to comment (5):

Thanks for your good comment. We attribute the sign change of the Seebeck coefficient to phonon drag. Please see more details in the response to comment (4). As for the reason for why the electrons have higher mobility than the holes, we at present can only suppose that it may be because of the slightly higher effective masses of the hole pockets than those of the electron pockets, according to the SdH analysis. We noticed that the difference between the effective masses is not large, indicating that there can also be other reasons, and it would be very interesting for further studies.

REVIEWERS' COMMENTS

Reviewer #1 (Remarks to the Author):

The authors have addressed reviewers comments reasonably well, thus I am fine with the publication of this work in Nature Communications.

Reviewer #2 (Remarks to the Author):

The authors have addressed all of my comments and the revision is satisfactory. Now I can recommend the publication of this paper in Nat. Commun.

Additional minor comment:

There are several different cultures in the definition of the isothermal/adiabatic figures of merit for transverse thermoelectric conversion. On page 4 of the supplementary information, it is better to describe the definition of the figures of merit and constituent parameters with appropriate references.

Reviewer #3 (Remarks to the Author):

In this revision the authors have done a diligent job addressing my comments from the previous review, and I think the paper can be accepted to Nature Communications.

COMMENTS TO AUTHOR:

Reviewer #1: The authors have addressed reviewers comments reasonably well, thus I am fine with the publication of this work in Nature Communications.

Response: We greatly appreciate the reviewer's approval of this work.

Reviewer #2: The authors have addressed all of my comments and the revision is satisfactory. Now I can recommend the publication of this paper in Nat. Commun.

Additional minor comment:

There are several different cultures in the definition of the isothermal/adiabatic figures of merit for transverse thermoelectric conversion. On page 4 of the supplementary information, it is better to describe the definition of the figures of merit and constituent parameters with appropriate references.

Response: We greatly appreciate the reviewer's approval of this work. As for the minor comment, we have described the definition of the figure of merit in the supplementary information on page 4, with references as well.

“Using adiabatic resistivity, the obtained Nernst zT is the adiabatic figure of merit as defined in previous report.³” (supplementary, page 4)

3. C. F. Kooi, R. B. Horst, K. F. Cuff, S. R. Hawkins, Theory of the longitudinally isothermal Ettingshausen Cooler, *J. Appl. Phys.* **34**, 1735 (1963). (supplementary, reference, page 6)

Reviewer #3: In this revision the authors have done a diligent job addressing my comments from the previous review, and I think the paper can be accepted to Nature Communications.

Response: We greatly appreciate the reviewer for approving this work.